# Comparative Transcriptomic Analysis of the Pituitary Gland between Cattle Breeds Differing in Growth: Yunling Cattle and Leiqiong Cattle

**DOI:** 10.3390/ani10081271

**Published:** 2020-07-25

**Authors:** Xubin Lu, Abdelaziz Adam Idriss Arbab, Zhipeng Zhang, Yongliang Fan, Ziyin Han, Qisong Gao, Yujia Sun, Zhangping Yang

**Affiliations:** 1College of Animal Science and Technology, Yangzhou University, Yangzhou 225009, China; dx120180094@yzu.edu.cn (X.L.); arbabtor@yahoo.com (A.A.I.A.); zzp19930311@126.com (Z.Z.); dx120170088@yzu.edu.cn (Y.F.); mx120170661@yzu.edu.cn (Z.H.); 18305182715@163.com (Q.G.); 2Joint International Research Laboratory of Agriculture and Agri-Product Safety, Yangzhou University, Yangzhou 225009, China; 007060@yzu.edu.cn

**Keywords:** Yunling cattle, Leiqiong cattle, transcriptome, pituitary gland, growth, hormones

## Abstract

**Simple Summary:**

Growth and development are very important in agricultural animals because they remarkably affect the value of the livestock. Although both Yunling cattle and Leiqiong cattle belong to the same subspecies, *Bos taurus indicus* (also known as the zebu cattle), they show extreme difference in growth rate from birth to adulthood. The pituitary gland is a key endocrine organ that regulates the secretion of growth-related hormones. To identify the core genes controlling growth by regulating the expression of hormones, in this study, the contents of six growth-related hormones were detected in the plasma of the two breeds, and transcriptome sequencing was performed to screen the differentially expressed genes related to the secretion of these hormones. Our study revealed that the overall gene expression levels are quite similar between the two breeds; eight new candidate genes were detected that may regulate the growth of cattle by regulating the secretion of growth-related hormones. Our findings will help researchers further understand the mechanism of the pituitary gland’s influence on growth and development in cattle. The results can contribute to the artificial selection and development of the beef cattle industry.

**Abstract:**

The hypothalamic–pituitary–thyroid (HPT) axis hormones regulate the growth and development of ruminants, and the pituitary gland plays a decisive role in this process. In order to identify pivotal genes in the pituitary gland that could affect the growth of cattle by regulating the secretion of hormones, we detected the content of six HPT hormones related to growth in the plasma of two cattle breeds (Yunling and Leiqiong cattle, both also known as the zebu cattle) with great differences in growth and compared the transcriptome data of their pituitary glands. Our study found that the contents of GH, IGF, TSH, thyroxine, triiodothyronine, and insulin were significantly different between the two breeds, which was the main cause of the difference in growth; 175 genes were identified as differentially expressed genes (DEGs). Functional association analyses revealed that DEGs were mainly involved in the process of transcription and signal transduction. Combining the enrichment analysis and protein interaction analysis, eight DEGs were predicted to control the growth of cattle by affecting the expression of growth-related hormones in the pituitary gland. In summary, our results suggested that *SLC38A1*, *SLC38A3*, *DGKH*, *GNB4*, *GNAQ*, *ESR1*, *NPY*, and *GAL* are candidates in the pituitary gland for regulating the growth of Yunling and Leiqiong cattle by regulating the secretion of growth-related hormones. This study may help researchers further understand the growth mechanisms and improve the artificial selection of zebu cattle.

## 1. Introduction

Growth and development are extremely important in beef cattle production because they significantly impact farmers’ income, as well as adjust and improve the dietary structure and people’s living standards. A large amount of cattle research has focused on the molecular regulation mechanisms related to growth and development [1,2].

The pituitary is the most critical and complex endocrine gland. Located on the underside of the brain and connected to the hypothalamus, it has been demonstrated to regulate growth and development by secreting a variety of hormones such as growth hormone (GH) and thyroid-stimulating hormone (TSH) [3,4,5]. Changes in the level of GH secreted by the pituitary gland can directly affect insulin-like growth factor-I (IGF-1) secretion and regulate postnatal growth in cattle [4]. Aside from causing IGF-1 release, GH antagonizes insulin action, which could affect the development of lean tissue by nutritional distribution [6]. TSH is mainly responsible for regulating the proliferation of thyroid cells and the synthesis and secretion of thyroid hormones (TH) and affects the growth and metabolism of cattle. The production of cattle therefore involves GH, TSH, IGF-1, and insulin because they act together on individual animals [7]. Many genes that participate in the process have been identified, such as *MYC*, *PPARG*, *GSK3B*, *TG*, *STAT5b*, and *IYD* [8,9]. However, the growth and development of biological organisms is very complicated, and there are still many genes involved that need to be identified [10,11].

As a new fast and efficient transcriptome research method, high-throughput RNA sequencing (RNA-seq) can help researchers to reveal the genetic mechanisms of animals and improve the production of animal products [12]. Yunling and Leiqiong are beef cattle breeds in the south of China. Yunling cattle have been cross-bred for almost 40 years using three breeds of Brahman (1/2), Murray Grey (1/4), and Chinese native cattle (1/4). Adult Yunling cattle generally weigh more than 500 kg. With the characters of enhanced growth, high meat production rate, and good reproductive capacity under high temperature and high humidity conditions, Yunling cattle have become an important source of beef production in China [13]. Leiqiong cattle are a Chinese native breed with more than 2000 years of breeding history in China’s southernmost regions such as Guangdong province and Hainan province. Research has shown that Leiqiong cattle may originate from *Bos indicus* [14]. Leiqiong cattle have strong limbs, firm hooves, and thin and elastic skin, and along with the characteristics of heat resistance and strong tick resistance, Leiqiong cattle are often used for labor in rural areas of China. However, the growth and meat production rate are usually low, and adult Leiqiong cattle generally weigh around 200 kg [15]. Yunling and Leiqiong cattle breeds all belong to the same subspecies, *Bos taurus indicus* (also known as the zebu cattle), but show great differences in growth rate from birth to adulthood. Our research will reveal the effect of the pituitary gland on the growth of zebu cattle at the molecular level.

To further understand the molecular interaction in the pituitary glands of cattle and identify pivotal genes that control growth by affecting the secretion of growth-related hormones, we detected the content of six hypothalamic–pituitary–thyroid axis (HPT) hormones, GH, IGF-1, TSH, triiodothyronine, thyroxine, and insulin, in the plasma of Yunling and Leiqiong cattle breeds and performed RNA-seq technology on the pituitary glands of these two breeds.

## 2. Materials and Methods

### 2.1. Ethical Statement

This research was conducted in strict accordance with the Regulations of the Administration of Affairs Concerning Experimental Animals (China, 2004) and approved by the Institutional Animal Care and Use Committee of School of the Yangzhou University Animal Experiments Ethics Committee (License Number: SYXK (Su) IACUC 2012-0029). All animals were raised and slaughtered according to the Standards for the Administration of Experimental Practices (Jiangsu Province, China, 2008).

### 2.2. Animals and Sample Collection

Three Yunling bulls and three Leiqiong bulls of 12 months old were selected and raised at the Academy of Grassland and Animal Science (Yunnan Province, China) for six months. All the cattle were born on the same day, and there was no close family relationship (at least three generations back) between them. All cattle were not castrated in this study, and they were all in the stage of sexual maturity throughout the 6-month experiment. Before the experiment, they were provided enough feed and all of them showed normal growth (no disease or nutritional deficiencies). Then the cattle were kept in one cowshed and fed the same diet, which consisted of 56% corn, 6% soybean meal, 8% cottonseed meal, 1.2% vegetable oil, 13% rapeseed dregs, 12.5% wheat bran, 1.3% CaHPO_4_, 0.5% NaCl, 0.5% NaHCO_3_, and 1.0% premix. All six bulls were provided enough feed and could access feed throughout the day during the experiment. The average birth weight of Yunling and Leiqiong cattle was 28.37 ± 1.32 kg and 15.02 ± 2.14 kg, respectively. The average weight of Yunling cattle at the age of 12 months was 292.21 ± 8.65 kg, while the average weight of Leiqiong cattle was 132.72 ± 5.11 kg. At the time of slaughter (18 months), the average body weight of the two breeds was 496.8 ± 14.03 kg (Yunling) and 193.60 ± 11.21 kg (Leiqiong), and the average daily gain of the Yunling and Leiqiong cattle breeds was 1.13 ± 0.24 kg and 0.34 ± 0.08 kg during the six-month experiment, respectively. The daily gain and the weight at birth, at 12 months and at slaughter, of the two breeds were analyzed using independent samples *t*-test performed in SPSS (v. 26.0), and all have significant difference (*p* < 0.01). At the age of 18 months, about 2 h after they had last been fed, they were chemically anesthetized and humanely slaughtered without any pain at Jinggu Zhaohui Slaughterhouse (Yunnan Province, China). After the cattle were slaughtered, the entire pituitary fossa in the lower thalamus was obtained by cutting the skull. Then, the diaphragm on the surface of the pituitary fossa was carefully removed with tweezers. The pituitary glands were placed in RNase-free cryopreservation tubes and immediately placed in liquid nitrogen for subsequent analysis.

During the week before slaughter, after removing feed from the cattle for 2 h, blood samples from the two cattle breeds were collected from the jugular vein every afternoon. Samples were stored in sterile EDTA blood collection tubes (Becton, Dickinson and Company, New York, NY, USA) and immediately placed on ice packs and brought into the laboratory for refrigeration. Then the blood was centrifuged at 1015.30× *g* for 30 min to separate out the plasma, and the plasma was stored at −20 °C for further measurements.

### 2.3. Quantification of Growth-Related Hormones

According to the manufacturer’s instructions, six hormones from plasma samples, GH, IGF-1, TSH, triiodothyronine, thyroxine, and insulin, were quantified using bovine-specific ELISA kits; the catalog (CAT) number for the kits detecting GH, IGF-1, TSH, triiodothyronine, thyroxine, and insulin were *EY-ELISA-3428*, *EY-ELISA-0326*, *EY-ELISA-6368*, *EY-ELISA-7746*, *EY-ELISA-7747*, and *EY-ELISA-5325*, respectively. All six kits were procured from the Y-S Biotechnology Company (Shanghai, China). The optical density (OD) at 450 nm was measured by a Tecan Nano Quant ELISA reader (Infinite M200 PRO, Männedorf, Switzerland). The coefficient of variation (CV) of every hormone’s intra-assay was less than 10%. The average concentrations of the six hormones between the two breeds were analyzed using independent samples *t*-test performed in SPSS (v. 26.0).

### 2.4. RNA Extraction and Sequencing

The total RNA of each pituitary gland sample was extracted with Trizol reagent (15596026, Invitrogen, Carlsbad, CA, USA), following the manufacturer’s instructions. The RNA was quantified by 1% agarose gel electrophoresis and qualified by measuring the 260/280 nm absorbance on a NanoDrop ND-1000 spectrophotometer (IMPLEN, Westlake Village, CA, USA). Then the quality of RNA was evaluated by the RNA integrity number (RIN) in a 2100 Bioanalyzer (Agilent Technologies, Santa Clara, CA, USA). Only samples of high quality (RIN > 8) were used to construct the RNA-seq library.

All six samples’ cDNA libraries were obtained by a TruSeq Stranded mRNA LT Sample Prep Kit (RS-122-2101, Illumina, San Diego, CA, USA), and the polyadenylated mRNA was purified and concentrated with oligo (dT)-conjugated magnetic beads (Thermo Fisher, Waltham, MA, USA) and fragmented with the Elute, Prime, Fragment Mix (Illumina) at 95 °C. Then end repair and 5′ adaptor ligation were conducted [16]. After that, reverse transcription was carried out with RT primers, and end repair and 3′ end adenylation were performed. Then the cDNA was purified and amplified via 10 PCR cycles, and the obtained libraries were refined by Agencourt AMPure XP beads (Beckman, Miami, FL, USA), validated by the Agilent Bioanalyzer 2100 Bioanalyzer (Agilent, Santa Clara, CA, USA), and calibrated by the Qubit dsDNA HS Assay Kit (Q32854, Thermo Fisher Scientific, Waltham, MA, USA). Finally, the cDNA libraries were detected on an Illumina HiSeq 2500 using 2 × 150 bp paired-end sequencing, following the manufacturer’s instructions of the TruSeq SBS v3-HS Kit (fc-401-3002, Illumina, San Diego, CA, USA).

Raw reads were discarded if they contained more than two N bases; then the adaptors and the low-quality bases of sequences were removed, and reads less than 16 nt were discarded again by the FASTX-Toolkit (v. 0.0.13) [17]. Finally, the quality of the reads after cleaning was assessed by FastQC (v. 0.11.2): reads with a’ quality value of more than Q30 (correct rate more than 99.9%) were called clean reads and could be used in the next step [18].

### 2.5. RNA Seq Data Assembly and Functional Assignment

The reference genome (ftp://ftp.ncbi.nlm.nih.gov/genomes/all/GCF/002/263/795/GCF_002263795.1_ARS-UCD1.2/GCF_002263795.1_ARS-UCD1.2_genomic.fna.gz) and annotation file of zebu cattle (ftp://ftp.ncbi.nlm.nih.gov/genomes/all/GCF/002/263/795/GCF_002263795.1_ARS-UCD1.2/GCF_002263795.1_ARS-UCD1.2_genomic.gff.gz) were downloaded from the NCBI database. Then the clean reads were aligned against the reference genome by TopHat2 (v. 2.1.1) (mismatch is set to 2; the other parameters are the defaults) [19]. HTSeq (v. 0.11.2) was used to estimate the relative transcript abundance and the calculation method is fragments per kilobase per million (FPKM) [20].

The Bioconductor package DESeq2 (v. 1.6.2) was used to identify the differentially expressed genes (DEGs) between the pituitary glands of Yunling and Leiqiong cattle [21], and the fold changes (FC) along with the associated *p*-values of DEGs were obtained (Appendix A). The threshold of adjusted *p*-value < 0.05 and log_2_ |FoldChange| > 3 was used to define the DEGs.

### 2.6. Verification of Sequencing Data by RT-qPCR

In order to verify the accuracy of sequencing data and whether they could well represent the expression of genes, 10 randomly selected genes (*GHRHR*, *JAG1*, *FAM92B*, *SRMS*, *OPRK1*, *NELL2*, *FNIP2*, *SYTL2*, *CXCL14*, and *HTR3E*) were used to compare gene expression levels in the samples measured by RT-qPCR and the FPKM values in RNA-seq data.

TRIzol Reagent (15596-026, Invitrogen, Carlsbad, CA, USA) was used to extract the total RNAs from the same RNA-seq pituitary samples. RNA quality and quantity were assessed using a NanoDrop ND-1000 spectrophotometer (IMPLEN, Munich, Germany) and evaluated by the RNA integrity number (RIN) in a 2100 Bioanalyzer (Agilent Technologies, Santa Clara, CA, USA). Then the PrimeScript RT reagent Kit (RR047A, Takara, Dalian, China) was used to reverse transcribe the total RNA of each sample into cDNA in accordance with the manufacturer’s instructions. Beacon designer (v. 8.02) [22] was used to design the primers of the selected genes, and all the primers were synthesized by TsingKe Biotech (Nanjing, China). The real-time PCR reactions were performed with SYBR Premix Ex TaqTM II kit (RR820A, Takara, Dalian, China) on a Bio-Rad CFX96 real-time PCR detection system (Bio-Rad, Hercules, CA, USA) in triplicate, and three housekeeping genes, *RPL4*, *GAPDH*, and *RPL11,* were selected as internal control genes. The geometric mean of the internal control genes was used to normalize the expression data. The 2^−ΔΔCt^ method was used to calculate the expression of each gene. The details of the primer sequences are given in Appendix A.

To evaluate the accuracy of the sequencing data and the reproducibility across replicates, the correlation coefficient (R^2^) between the expression levels of the 10 selected genes measured by FPKM, and the RT-qPCR was analyzed by a Pearson test conducted on R statistical software program (v. 3.6.2) [23].

### 2.7. Functional Gene Annotation

The Bioconductor package org.Bt.eg.db (v. 3.10.0) was used to change the bovine official gene symbols to bovine Ensembl gene IDs; then, the Bioconductor package cluster profiler (v. 3.14.3) was used to carry out the gene ontology (GO) terms analysis and Kyoto Encyclopedia of Genes and Genomes (KEGG) pathways analysis in the R statistical software program (v. 3.6.2) [23]. If the *p*-value was less than 0.05, the biological processes and pathways enriched by DEGs were considered to be significantly different.

### 2.8. PPI Network Construction

In order to discover the DEGs controlling the growth of cattle by regulating the secretion of the growth-related hormones studied in this article, the six encoding proteins genes involved in regulating the secretion of these hormones, *GH*, *IGF-I*, Thyrotropin subunit beta precursor (*TSHB*), Thyroid hormone receptor alpha (*THRA*), Thyroid hormone receptor beta (*THRB*), and Insulin (*INS*), were selected, and the differences in them between Yunling and Leiqiong cattle were calculated by the independent samples *t*-test method in SPSS (v. 26.0).

A protein–protein interaction (PPI) network was created by Cytoscape (v. 3.8.0) software [24]. DEGs encoding proteins that may have a relationship with the above six genes’ encoding proteins were predicted by the String (v. 11.0) database [25], and their interacting partners were subsequently visualized in Cytoscape (v. 3.8.0) [24].

## 3. Results

### 3.1. Growth-Related Hormones’ Expression Analysis

To further understand the mechanism of growth and development of zebu cattle, the contents of growth-related hormones including GH, IGF-1, TSH, triiodothyronine, thyroxine, and insulin were detected; the results are shown in Figure 1. From the results, we know that the concentrations of the six hormones are all significantly different in the plasma of Yunling cattle and Leiqiong cattle (*p* < 0.05). Of these, the concentrations of GH, IGF-1, TSH, triiodothyronine, and thyroxine were significantly higher in Yunling cattle, and the concentration of insulin was higher in Leiqiong cattle (Figure 1).

### 3.2. Summary of Mapping Statistics

On average, 60,936,319 raw reads for each sample was generated by sequencing the pituitary glands of Yunling and Leiqiong cattle. After cutting the adapter sequences of reads and discarding the low-quality reads, 91.12% to 94.80% of the raw reads were retained, of which 84.65% to 93.51% of the reads had an error rate less than 0.1% (Q30). Among the reads with a quality value greater than Q30, 85.25% and 71.42% of reads from Yunling and Leiqiong cattle, respectively, were mapped to unique bovine genomic locations (Table 1). It is shown in the information of alignment (Figure 2a) that most of the unique reads (on average, 78.63% in Yunling cattle and 78.37% in Leiqiong) were mapped to the exons of genome, while just 11.30% to 9.60% were mapped to the introns and 10.07% to 12.03% were mapped to the intergenic regions.

### 3.3. Characteristics of Expression Profile Data

In total, 16,009 annotated genes with FPKM higher than 0.1 were identified (Appendix A). Of these, 15,682 were co-expressed between Yunling and Leiqiong pituitary gland samples, while only 131 (0.82%) genes were expressed only in Yunling cattle and 196 (1.22%) only in Leiqiong cattle (Figure 2b).

The expression distribution of all detected genes was compared between Yunling and Leiqiong cattle, and it was shown that the vast majority of genes’ FPKM were between 1 and 100 (79.22% in Yunling cattle and 76.39% in Leiqiong cattle); only a few genes’ FPKM were larger than 100 (3.01% in Yunling cattle and 3.31% in Leiqiong cattle). It is also interesting to note that the number of genes of the two breeds, divided according to the order of magnitude of FPKM, was quite similar (Figure 3a). In addition, the FPKM values of the six selected housekeeping genes, including hypoxanthine phosphoribosyl transferase 1 (*HPRT1*), ribosomal protein L4 (*RPL4*), *GAPDH*, ribosomal protein L11 (*RPL11*), ribosomal protein S27a (*RPS27A*), and ribosomal protein L13A (*RPL13A*), were analyzed by independent samples *t*-test performed in SPSS (v. 26.0), and there was no significant difference (*p* > 0.05) between the two cattle breeds. In summary, the overall expression of the detected genes in the pituitary glands of the Yunling and Leiqiong cattle groups was very similar (Figure 3b).

### 3.4. Validation of Samples’ Reproduction and RNA-seq Data’s Accuracy

To estimate the reproducibility between samples, a principal component analysis (PCA) was conducted on R statistical software program (v. 3.6.2) [23] based on the FPKM value of the co-expressed genes. Principal component 1 (PC1) and principal component 2 (PC2) together explained 73.4% of the total variance (PC1 accounted for 54.8% and PC2 accounted for 18.6%). It is shown that samples from the Yunling and Leiqiong cattle groups can be clearly separated and gather within groups, which indicates that our samples are reproducible in each group (Figure 4a).

To quantitatively determine the reliability of RNA-Seq data, we performed RT-qPCR on 10 randomly selected genes, *FNIP2*, *FAM92B*, *JAG1*, *GHRHR*, *OPRK1*, *CXCL14*, *NELL2*, *SYTL2*, *HTR3E*, and *SRMS*, to test their expression levels. The average RIN value derived from the six samples was 8.50 ± 1.10, which shows that the quality meets the sequencing requirements. For these 10 selected genes, a linear regression analysis was performed using log_2_ FC obtained by the RNA-seq and RT-qPCR, which revealed a high correlation (R^2^ = 0.88219) and confirmed the reliability of the current RNA-seq data in this study (Figure 4b). The inconsistency regarding the ratio may be due to the difference in the algorithms between the two methods.

### 3.5. Identification of DEGs

A total of 175 DEGs were identified (adjusted *p*-value < 0.01 and log_2_ |FoldChange| > 3; Appendix A) through statistical analysis of the RNA-seq data of Yunling and Leiqiong cattle pituitary gland samples. Among these DEGs, there were 31 genes that were more highly expressed in the Yunling cattle group and 144 genes in the Leiqiong cattle group. A volcano plot was constructed with the data of adjusted *p*-value and log_2_ FC of the DEGs in these two groups (Figure 5). The details of the DEGs, including the name of each gene, the log_2_ fold change, adjusted *p* values, and the variation in gene expression differences (UP or DOWN) between the Yunling and Leiqiong cattle groups are presented in Appendix A.

### 3.6. Enrichment Analysis of DEGs

In order to better investigate the function of the 175 DEGs in the pituitary glands of cattle, we performed gene ontology (GO) analysis (Appendix A) and Kyoto Encyclopedia of Genes and Genomes (KEGG) pathway analysis (Appendix A) on these genes at the DAVID (https://david.ncifcrf.gov) [26]. The top 20 enriched GO biological process terms (*p* < 0.05) were mainly functionally related to the transcription process, including transcription, DNA-templated, positive regulation of transcription from RNA polymerase II promoter, negative regulation of transcription from RNA polymerase II promoter and negative regulation of transcription, DNA-templated, and the genes in these processes accounted for 34.85% of all the genes enriched in the top 20 GO biological process terms. Followed by the signal transduction such as small-GTPase-mediated signal transduction, intracellular signal transduction, neurotrophin TRK receptor signaling pathway, and synaptic transmission and the genes in these processes accounted for 21.20% of all the genes enriched in the top 20 GO biological process terms (Figure 6).

The KEGG analysis results (*p* < 0.05) listed in Table 2 reveal that the enrichments were mainly related to lipid and protein metabolism, such as Glycerophospholipid metabolism, Alanine, aspartate and glutamate metabolism, Mucin type O-glycan biosynthesis, Thyroid hormone signaling pathway, Endocrine and other factor-regulated calcium reabsorption, and Glutamatergic synapse.

### 3.7. PPI Network Analysis

All six genes involved in regulating the secretion of the six hormones studied in this article were significantly different (*p* < 0.05) in the Yunling and Leiqiong cattle pituitary gland samples. More specifically, the expression of *GH*, *IGF-I*, *TSHB*, and *THRB* was significantly higher in Yunling cattle, while the expression of *THRA* and *INS* was significantly higher in the Leiqiong cattle pituitary gland (Figure 7a). Fifteen DEGs’ encoding proteins were predicted to have a relationship with these genes’ encoding proteins, and the PPI network of them is shown in Figure 7b.

## 4. Discussion

The research on growth and development has important significance for the feeding and management of beef cattle. It is well known that growth is a highly complex and delicate physiological process regulated by nerves and body fluids. Growth axis hormones and their receptors play a significant role in regulating growth, and the pituitary gland is a principal endocrine organ that participates in this process [26,27]. In order to study the growth of cattle, many tissues have already been subjected to high-throughput sequencing, such as the longissimus dorsi muscle, fat, rumen tissues, and so on [28,29,30]. However, research on the effects of gene expression in the pituitary gland on growth is still limited. For the sake of better understanding the molecular mechanisms that control growth in the pituitary glands of cattle, we detected the content of six growth-related hormones in the plasma of two zebu cattle breeds (Yunling and Leiqiong cattle) with significant differences in growth and analyzed the RNA-seq data of their pituitary glands.

The growth and development of ruminants are affected by many factors such as variety, endocrine status, nutrition, and environment. GH is essential to maintaining normal bone growth, including achieving vertical bone growth and bone hardness. In most cases, GH generally affects skeletal cells indirectly through IGF-1 [31]. Thyroxine and triiodothyronine have an accelerating effect on puberty growth in cattle, and the proper addition of thyroxine and triiodothyronine can promote cell growth [32]. It is also found that the injection of thyroxine and triiodothyronine can increase the concentration of IGF in the study of dairy cows [33]. The concentration of GH, IGF, TSH, thyroxine, and triiodothyronine are significantly higher in Yunling cattle than in Leiqiong cattle, which indicates that these hormones may promote the growth and development of beef cattle and might be the direct cause of differences in growth (Figure 1). Insulin participates in the regulation of carbohydrate and fat metabolism and controls the balance of blood sugar, which can prompt the liver and skeletal muscle cells to convert the glucose in the blood into original sugar—that is, converting the blood sugar into hepatic sugar and intracellular glucose and insulin can increase the synthesis of glycogen fatty acids and proteins in cells [34]. During refeeding and compensatory growth in cattle, the secretion of insulin is sharply enhanced and the plasma GH concentrations are high, which leads to increased fat deposition [35]. This is different in our present study because all the experimental cattle had sufficient food every day. In this study, the concentration of GH was higher in Yunling cattle but insulin was higher in Leiqiong cattle; this may be caused by the following three reasons: (1) the differences in breeds—Leiqiong cattle are known to be more irritable and have stronger disease resistance [15]; (2) Yunling cattle were still in a growing period at 18 months, whereas the Leiqiong cattle were at a later stage of growth and not growing as much as Yunling—a study has reported that the insulin inhibitory effect in the growth period of cattle is greater than in the later growth period [36]; (3) the live-weight of Yunling cattle was much higher than Leiqiong cattle; as body weight increases, the body becomes more insulin-resistant [37,38].

Another factor that might affect the secretion of the growth-related hormones in Yunling and Leiqiong cattle in this study might be the function of androgen. It was reported that androgens could increase the expression of *IGF-I* in satellite cells and promote myogenesis in humans [39]. The high concentration of androgen could promote the secretion of insulin by potentiating the insulinotropic action of glucagonlike peptide-1 (*GLP-1*) in the study of rats [40] but had no significant effect on the secretion of thyroxine and triiodothyronine [41]. Although all of the cattle were in the stage of sexual maturity throughout the 6-month experiment, the androgen concentration might differ between the two breeds and may have affected the secretion of growth-related hormones in this study, but this also needs to be confirmed by further experiments.

Although a vast majority of the unique reads were located in the exon region (on average: 78.50%), a considerable amount of them were still mapped to the intronic (10.45%) and intergenic (11.05%) regions (Figure 2a); in fact, it was revealed that in the intronic and intergenic regions, large amounts of noncoding RNAs could be transcribed from and perform essential molecular functions [42,43], so genome annotation work in cattle still needs to be constantly improved. Even if the breeds of Yunling and Leiqiong cattle are different, a majority of genes are expressed at quite similar levels between them (Figure 3a). This result was the same as in our previous research on Simmental and Wenshan cattle [44], which illustrates a possibility that, in beef cattle, the main variation in gene expression might be caused by differences in tissue rather than variety.

Ensuring the accuracy of data, including sampling and sequencing, is a prerequisite for subsequent analysis. In this study, the expression levels of the six random selected housekeeping genes were not significantly different between the two groups (Figure 3b), which illustrated the uniformity of sequencing data between samples. Two principal components that contributed the most to the overall variance of the six samples, PC1 and PC2, were extracted, and the pituitary gland samples of Yunling and Leiqiong cattle were clearly separated in the PCA figure plotted according to PC1 and PC2 (Figure 4a), which illustrated the repeatability of samples collection. (We speculated that PC1 and PC2 might highly be correlated with the expression of growth-related genes because the distribution of these six samples was roughly in accordance with the daily gain of the two breeds during the six-month experiment; for example, the pituitary gland samples of Yunling group and Leiqiong group were clearly separated might due to the average daily gain of Yunling and Leiqiong was significantly different, and the Yunling group was not as close in the PCA plot as the Leiqiong cattle group, which may be caused by the variance in daily gain in the Yunling cattle group being higher than in the Leiqiong cattle group.) The log_2_ FC of 10 randomly selected genes calculated by two methods, RT-qPCR and RNA-seq data (Figure 4b), were significantly correlated with each other, which illustrated the accuracy of sequencing data. The above all revealed that the accuracy of data was reliable in this study.

In the present study, a total of 175 DEGs were identified between the pituitary glands of Yunling and Leiqiong cattle (Figure 5). These genes are mainly involved in the process of transcription and signal transduction, such as DNA-templated, positive regulation of transcription from RNA polymerase II promoter, negative regulation of transcription from RNA polymerase II promoter, negative regulation of transcription, DNA-templated, small GTPase mediated signal transduction, intracellular signal transduction, neurotrophin TRK receptor signaling pathway, and synaptic transmission (Figure 6). The process of transcription is an important way for genes to participate in the regulation of the body, and the efficiency of the process has an important impact on biological growth and development [45,46]. Studies have reported that the GO term small GTPase-mediated signal transduction can act upstream of mTOR to regulate the growth of *HEK293* cells [47], and the mammalian target of Rapamycin (mTOR) signaling pathway has recently been examined in ovarian follicles in mice, where it regulates granulosa cell proliferation and differentiation [48]. Via the neurotrophin TRK receptor signaling pathway, the Akt and MAPK pathways can control and transmit signals of cell growth and development [49]. Therefore, we infer that differences in transcription levels in the pituitary gland are one of the reasons for the growth levels in Yunling cattle being higher than in Leiqiong cattle.

DEGs are enriched in the 13 KEGG pathway, and some of them are related to tissue growth by regulating growth-related hormones (Table 2). Glutamatergic synapse can develop the growth neurons and promote the construction of cortical networks by affecting the expression of Growth hormone secretagogue receptor (*GHSR*) [50]. The Thyroid hormone signaling pathway plays a key role in muscle formation and fat metabolism in pigs by regulating thyroid hormones [51]. Alanine, aspartate, and glutamate metabolism can regulate pancreas development, muscle formation, and post-embryonic development in cattle [52]. The Glycerophospholipid metabolism pathway can inhibit insulin signaling and affect lipid production and metabolic processes in dairy cattle [53]. In studies of mice and poultry, it was also found that the Neuroactive ligand–receptor interaction pathway can regulate the growth and sexual maturity of individuals by affecting the release of growth hormone, thyroid hormone, and insulin [54,55]. Besides these pathways, many genes have been found to be candidate genes in the pituitary gland that may influence the growth of cattle, such as *SLC38A1*, *SLC38A3*, *DGKH*, *GNB4*, *GNAQ*, *ESR1*, *NPY*, and *GAL*.

Solute carrier family 38 member *1* (*SLC38A1*) and solute carrier family 38 member 3 (*SLC38A3*) are members of solute carrier family 38; they are amino acid transporter proteins that increase with age during growth. Interestingly, *SLC38A1* was abundantly expressed in the Yunling cattle’s pituitary glands, and *SLC38A3* was abundantly expressed in the Leiqiong cattle’s pituitary glands. In our previous study, we also found that *SLC38A4* was abundantly expressed in the Simmental cattle pituitary gland compared with Wenshan cattle [56]. There are many reports to prove that *SLC38A1*, *SLC38A3*, and *SLC38A4* are involved in mammalian growth regulation, and *SLC38A1* and *SLC38A4* are primarily implicated in growth control of the embryo and can positively regulate prenatal growth by regulating the release of insulin [57]. Downregulation of *SLC38A3* can reduce amino acid availability and impair growth in mice [58]. These results demonstrate that solute carrier family 38 genes have crucial functions in secretory pathways and could regulate the growth and development of beef cattle in the pituitary gland, but their specific responsibilities are different.

Diacylglycerol kinase eta (*DGKH*) was proposed as a regulator protein of the *Ras*/*Raf*/*MEK*/*ERK* signaling cascade, which controls a vast number of growths regulating processes in cells, including proliferation, transformation, and differentiation [59]. A study in German Holstein cattle also showed that *DGKH* could affect the signal transduction process of GnRH (gonadotropin-releasing hormone) and regulated divergent growth at the onset of puberty [60]. However, there is no relevant research report on beef cattle.

G protein subunit alpha q (*GNAQ*) gene is one of the candidate genes that relate to the growth of cattle. Among the 13 KEGG pathways that are differentially expressed, *GNAQ* appears in six pathways (Glutamatergic synapse, Endocrine and other factor-regulated calcium reabsorption, Circadian entrainment, Dopaminergic synapse, Gap junction, and Spinocerebellar ataxia). By knocking down the *GNAQ* gene in the mouse pituitary gland, the postnatal proliferation of pituitary somatotroph cells was strongly impaired, and plasma growth hormone (GH) levels were reduced to 15%. The hypothalamic levels of GH-releasing hormone (GHRH), an important stimulator of somatotroph proliferation, were strongly decreased, which causes somatotroph hypoplasia, dwarfism, and anorexia in mice [61]. In a study of Kazakh sheep, it was also demonstrated that appropriate folate concentrations could promote *GANQ* promoter methylation, which in turn affected GnRH secretion and the growth of sheep [62]. It has also been suggested that, in humans, a primary effect of *GNAQ* variance on *cAMP* levels in the thyroid would affect the production of T4 and T3 and feedback to alter TSH release by the pituitary gland [63]. It can be inferred that *GNAQ* participates in the expression process of multiple classic pituitary hormones in the mammalian pituitary, and its expression has an important regulatory effect on growth and development. However, currently there is no relevant research available on the *GNAQ* gene in cattle, so conducting in-depth research on the effect and mechanism of *GNAQ* in the pituitary gland is of great significance to the development of beef cattle.

Studies have also shown that the mutation of *GNB4* can affect human muscle growth [64]. In breast cancer samples, overexpression of *ESR1* can promote cell growth by enhancing the *IGF* signaling pathway [65]. *NPY* and *GAL* genes could regulate the feeling of hunger by influencing the secretion of insulin and GH [66].

The transcription level (measured by FPKM value) of the genes involved in regulating the above six hormones, including *GH*, *IGF-I*, *TSHB*, *THRA*, *THRB*, and *INS*, were also significantly different with Yunling and Leiqiong group (Figure 7a). The transcription levels of these genes were consistent with the hormone concentration in blood of the two breeds (Figure 1), except *THRA,* which was higher in the Leiqiong than the Yunling group. Although the action of triiodothyronine and thyroxine was predominantly mediated by thyroid hormone receptors *THRA* and *THRB*, the effect of *THRA* was very low, and *THRA* mutations had little effect on the HPT axis [67,68]. Perhaps the differences in triiodothyronine and thyroid hormones in the blood were mainly caused by the different transcription levels of *THRB* in this study. The study also showed that mRNA transcript abundances can only explain about 40% of the variance in protein levels [69]. Future experiments are necessary to verify the relationships between these genes and their impacts on protein levels.

Fifteen DEGs’ encoding proteins were predicted to have a regulation relationship with the growth axis hormones studied in this research (GH, IGF-1, TSH, triiodothyronine, thyroxine, and insulin). Combined with the KEGG results, seven genes, *GNB4*, *GNAQ*, *ESR1*, *NPY*, *GAL*, *KLK1*, and *CLOCK*, were again detected in the PPI network results (Figure 7b) and *GNB4*, *GNAQ*, *ESR1*, *NPY*, and *GAL* in the pathways we have stated that affect hormone regulation. Therefore, combining the results of KEGG and PPI, it is possible that in Yunling and Leiqiong cattle, differential expression of *SLC38A1*, *SLC38A3*, *DGKH*, *GNB4*, *GNAQ*, *ESR1*, *NPY*, and *GAL* caused the secretion of growth-related hormones in the pituitary gland to be different, and eventually caused a variation in growth.

Determination of functional genes is helpful to the domestication and selective breeding of livestock species [70]. Artificial selection could target a lot of functional genes that could contribute to the improvement of traits that humans need, such as meat production, and increase the frequency of these favorable genes [71]. Yunling cattle were bred by cross-breeding (Brahman (1/2), Murray Grey (1/4), and Chinese native cattle (1/4)) for almost 40 years, and their growth and development were greatly improved, which was also mainly due to the favorable accumulation of gene frequency by artificial selection [72]. The live weight of a Brahman bull can reach 900 kg, which is much higher than Chinese native cattle [73], and Brahman may provide excellent genetic information for the growth of Yunling cattle breed. Studies have shown that the higher expression of *NPY* in hypothalamic could increase the release of NPY, which significantly increased the weight of Brahman cattle [74]. The expression changes of *ESR1* in pituitary gland could regulate the tissue and organ development of the pubertal Brahman cattle [75]. Then, it was supposed that in the process of cultivating and artificial selection of Yunling cattle during generations, the frequency of the eight genes, *SLC38A1*, *SLC38A3*, *DGKH*, *GNB4*, *GNAQ*, *ESR1*, *NPY*, and *GAL*, that were identified may be related to the growth of Yunling cattle in this study, which would gradually be changed to meet the needs of meat production and adapting to various environments. Further experiments like polymorphism detection of these genes for each generation of Yunling cattle might more deeply explain the hypothesis.

## 5. Conclusions

In conclusion, our study provides a basic exploration of the genes expressed in the pituitary glands of zebu cattle and identifies key genes that could regulate the secretion of growth-related hormones. The significant differences in the concentrations of GH, IGF, TSH, thyroxine, triiodothyronine, and insulin in the plasma might be the main cause of the growth differences in cattle. Overall expression distribution of the identified genes was quite similar between the Yunling and Leiqiong cattle, and a total of 175 genes were identified as differentially expressed genes, which are mainly involved in the process of transcription and signal transduction. We also identified eight genes, *SLC38A1*, *SLC38A3*, *DGKH*, *GNB4*, *GNAQ*, *ESR1*, *NPY*, and *GAL*, that may have a critical role in regulating the growth and development of zebu cattle. These genes may affect the secretion of growth-related hormones in the pituitary gland, which causes the differences in growth of the two breeds. Our findings will help researchers further understand the mechanism of the pituitary gland’s influence on growth and development in cattle, and the results may contribute to the artificial selection and development of the beef cattle industry.

## Figures and Tables

**Figure 1 animals-10-01271-f001:**
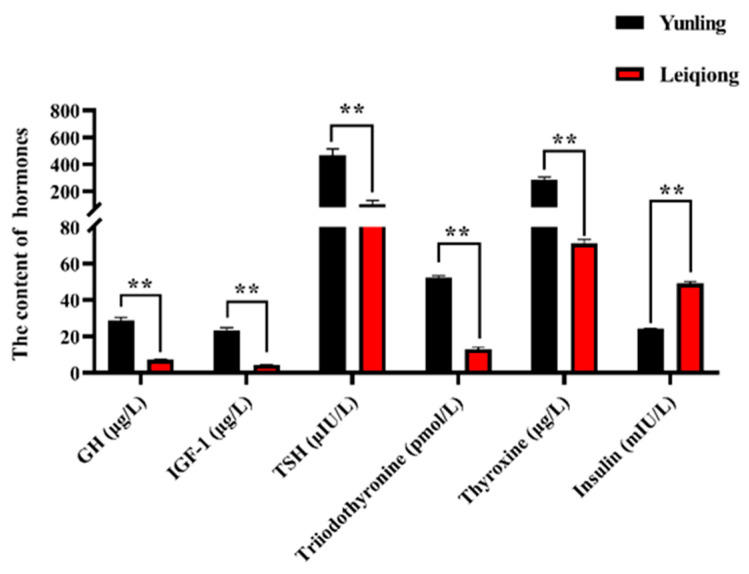
The concentration of hormones in the plasma of the two breeds (the black columns are Yunling cattle and the red columns are Leiqiong cattle, ** *p* < 0.05).

**Figure 2 animals-10-01271-f002:**
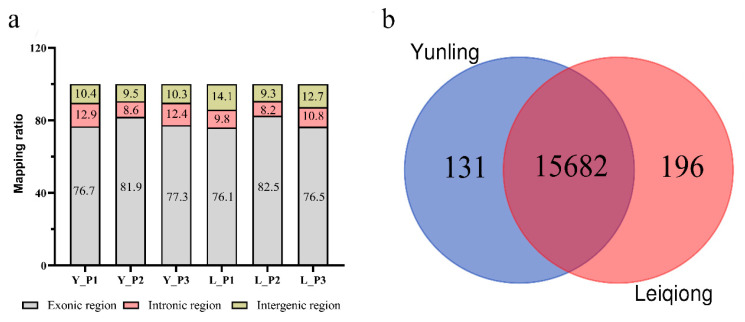
Genome alignment profiles. (**a**) The percentage of reads mapping to indicate genomic regions: exonic (gray), intronic (red), intergenic (yellow). (**b**) Venn diagram depicting the co-expressed (purple region) and uniquely expressed genes (blue region and red region) discovered in the pituitary glands of the two cattle breeds.

**Figure 3 animals-10-01271-f003:**
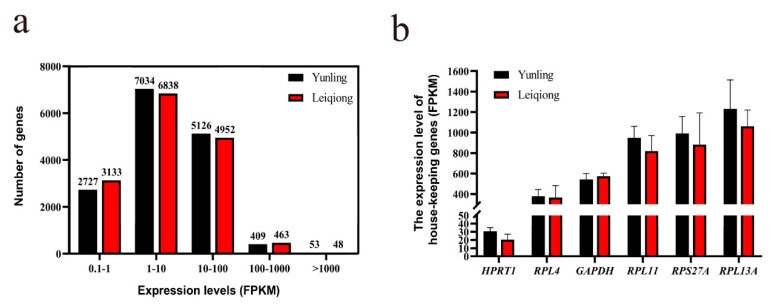
Basic analysis of RNA-Seq data. (**a**) Distribution of the numbers of detected genes at different expression levels measured by FPKM (mean ± SEM); (**b**) expression levels of six housekeeping genes in the pituitary glands of the two breeds measured by FPKM (mean ± SEM). FPKM, fragments per kilobase per million.

**Figure 4 animals-10-01271-f004:**
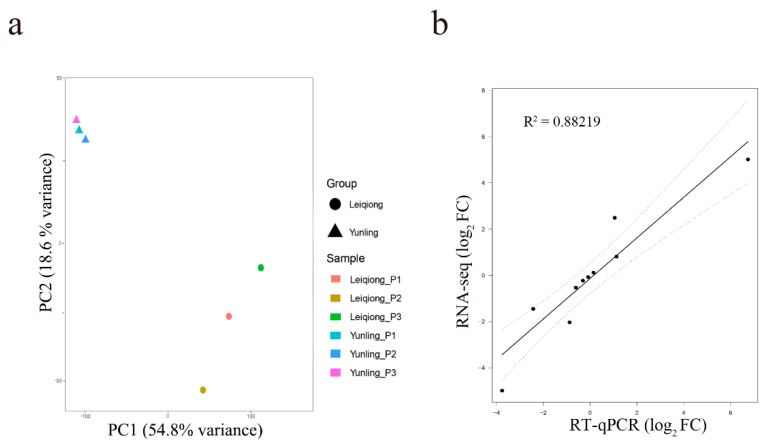
Validation of the accuracy of data. (**a**) Principal component analysis (PCA) results of the profiling data (circles represent Yunling cattle and triangles represent Leiqiong cattle); (**b**) correlation between RNA-seq and qRT-PCR data. The correlation coefficient between RNA seq (*y*-axis) and RT-qPCR (*x*-axis) data (log_2_ fold change) was analyzed by a Pearson test. R^2^ is 0.88219 with a statistical significance of *p* < 0.05.

**Figure 5 animals-10-01271-f005:**
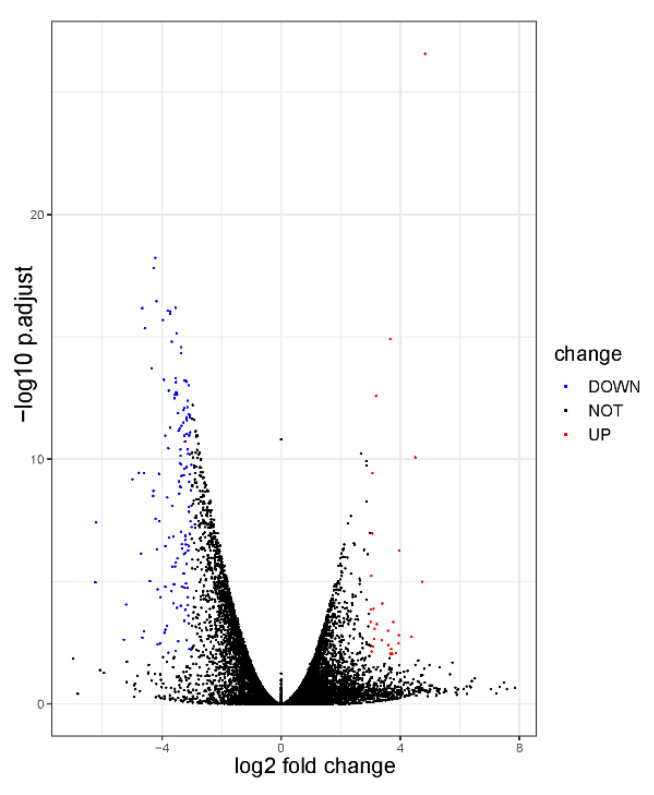
Volcano plot of the differentially expressed genes (DEGs) between Yunling and Leiqiong cattle pituitary glands. The red dots represent the upregulated transcripts (log_2_ fold change > 3, false discovery rate (FDR) < 0.01), and the blue dots represent the downregulated transcripts (log_2_ fold change < −3, false discovery rate (FDR) < 0.01).

**Figure 6 animals-10-01271-f006:**
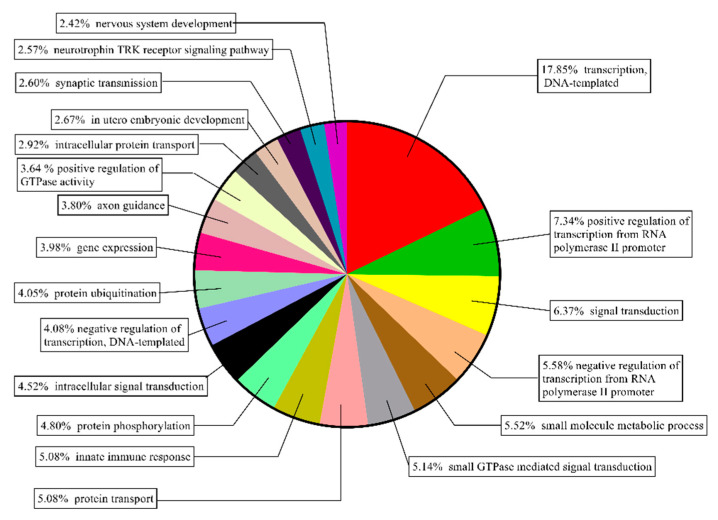
Gene ontology analysis (biological process) of differentially expressed genes between Yunling and Leiqiong cattle pituitary glands. (The top 20 enriched gene ontology (GO) biological process terms (*p* < 0.05. Each color represents a term.)).

**Figure 7 animals-10-01271-f007:**
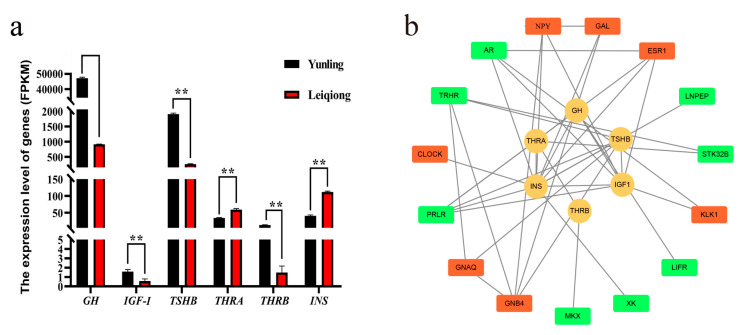
Expression levels of hormone-regulating genes and protein–protein interaction (PPI) network of DEGs with them. (**a**) Expression analysis of hormone-regulating genes, measured by FPKM (mean ± SEM, ** *p* < 0.05); All genes have significant expression differences between the two groups, of which *GH*, *IGF-I*, *TSHB,* and *THRB* were higher in Yunling group and *THRA* and *INS* were higher in Leiqiong group; FPKM, fragments per kilobase per million; (**b**) PPI network of DEGs with these hormone-regulating genes (the nodes in green are DEGs that did not appear in significantly different KEGG pathways (*p* < 0.05); the nodes in red are DEGs that also appeared in significantly different KEGG pathways (*p* < 0.05); the nodes in yellow are six hormones-regulating genes in this research).

**Table 1 animals-10-01271-t001:** Summary of reads mapping to the bovine transcriptomes.

Sample	Raw Reads	Clean Reads	Q30	Total Mapped	Multiple Mapped	Uniquely Mapped
Yunling_P1	57,697,726	53,001,188	90.00%	89.74%	3.86%	85.89%
Yunling_P2	58,487,994	53,293,322	89.72%	89.42%	4.81%	84.62%
Yunling_P3	60,623,110	55,260,350	89.82%	89.43%	4.20%	85.23%
Leiqiong_P1	65,019,640	59,273,088	84.65%	66.21%	2.25%	63.96%
Leiqiong_P2	63,584,756	60,275,154	93.51%	79.65%	4.09%	75.56%
Leiqiong_P3	60,204,686	56,734,662	92.89%	78.26%	3.53%	74.73%

**Table 2 animals-10-01271-t002:** Information on differentially expressed genes’ enrichment pathways.

Pathway	Description	Gene Name	*p*-Value
bta04724	Glutamatergic synapse	*SLC38A1* → *GNB4* → *GRIN2A* → *SLC38A3* → *GNAQ*	0.0011
bta04961	Endocrine and other factor-regulated calcium reabsorption	*ESR1* → *GNAQ* → *KLK1*	0.0050
bta04713	Circadian entrainment	*GNB4* → *GRIN2A* → *GNAQ* → *GUCY1A2*	0.0051
bta04919	Thyroid hormone signaling pathway	*ESR1* → *THRB* → *ITGAV* → *MED12L*	0.0091
bta04080	Neuroactive ligand–receptor interaction	*NPY* → *GRIN2A* → *GAL* → *THRB* → *TRHR*	0.0127
bta04614	Renin-angiotensin system	*KLK1* → *LNPEP*	0.0140
bta04728	Dopaminergic synapse	*GNB4* → *GRIN2A* → *GNAQ* → *CLOCK*	0.0140
bta00512	Mucin type *O*-glycan biosynthesis	*GCNT4* → *GALNT5*	0.0196
bta04540	Gap junction	*MAP3K2* → *GNAQ* → *GUCY1A2*	0.0247
bta04727	GABAergic synapse	*SLC38A1* → *GNB4* → *SLC38A3*	0.0254
bta00250	Alanine, aspartate, and glutamate metabolism	*AGXT2* → *GFPT1*	0.0273
bta05017	Spinocerebellar ataxia	*GRIN2A* → *GNAQ* → *KCND3*	0.0292
bta00564	Glycerophospholipid metabolism	*LPGAT1* → *ETNK1* → *DGKH*	0.0358

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
