# Peer review of "Comparative Transcriptomic Analysis of the Pituitary Gland between Cattle Breeds Differing in Growth: Yunling Cattle and Leiqiong Cattle"

_animals, 2020, doi:10.3390/ani10081271_

Round 1
Reviewer 1 Report
all of my original comments have been addressed and only have three minor comments on the revised manuscript:
Diet composition doesn’t add up to 100%
Line 258: 6 selected housekeeping genes, only three housekeeping genes are mentioned in methods section on line 185?
Supplementary tables S4 and S5 aren’t mentioned within the text?
Author Response
all of my original comments have been addressed and only have three minor comments on the revised manuscript:
Diet composition doesn’t add up to 100%
Response: Thanks for your reminder and we mistyped the content of NaCl of diet here, in fact the NaCl just occupied 0.5% of the diet, not 0.6% and we have corrected it, as is shown in line 101.
Line 258: 6 selected housekeeping genes, only three housekeeping genes are mentioned in methods section on line 185?
Response: Thanks for your comment. In fact, the 6 selected housekeeping genes in line 258 (now is in line 256) were used to verify the accuracy of the RNA-seq data by comparing their expression levels (FPKM) between the two groups, but for the RT-qPCR experiment, we only chosen three genes (RPL4, GAPDH, and RPL11) from the 6 genes as internal control genes in line 185 (now is in line 186). Maybe our statement is a little misleading and we change the sentence, as is shown in line 186-187.
Supplementary tables S4 and S5 aren’t mentioned within the text?
Response: Thanks for your reminder very much, and we have mentioned tables S4 and S5 in the manuscript. As is shown in line 303 and line 304, and line 531-532.

Reviewer 2 Report
The readability of the manuscript have been improved by the English editing service, however, I have some question to the authors described below.
- P1L35: What is the content of “nine DEGs” here (maybe eight)?
- P2L69: ~~ in China [13] Leiqiong ~~ → ~~ in China [13]. Leiqiong ~~
- P3L94-95: How about the impact of age (or growth stage) and sex hormone on the results in this study? The authors should discuss about these points in detail.
- P5L190: R2 generally means the coefficient of determination. Did the author want to show the value of R2? The authors must check this point.
- P5L203: DGE → DEG
- P8L270: How about the contribution rate of, and what is the biological meaning of each principal component? Discussion about these points seems to be better.
- P12L329-330: I recognize that genes THRA, and THRB encodes hormone receptor subunits, but not hormone itself. Therefore, I think this sentence seems incorrect.
- P13L363: I think the term “confirms” is too strong in this sentence.
- P13L387-388: I think the sentence “ which indicate that, ~~~ rather than variety.” seems to be too strong.
- P15L473: hypothalamic-pituitary-thyroid → HPT
Author Response
Please see the attached PDF file.

Reviewer 3 Report
The authors have revised the manuscript, titled “Comparative Transcriptomic Analysis of Pituitary Gland Between Cattle Breeds Differing in Growth: Yunling Cattle and Leiqiong Cattle” carefully based on the comments I gave earlier, not only improve many details and methods especially the feeding details of laboratory animals and the labeling of experimental reagents, but also give a reasonable answer to the statistical methods mentioned in this study. Moreover, there is a big improvement in the expression of English, no grammar and spelling errors are found.
The authors used reasonable research methods and analytical methods and presented 8 new candidate genes that regulate the growth of beef cattle by affecting hormone secretion. The study is of interest and convincing. The paper has certain novelty and advantages for the field research work of cattle growth and development, and has value for publishing in animals.
I think the manuscript has met the publishing requirements and I suggest it to be published.
Author Response
Please see the attached PDF file.

Reviewer 4 Report
The important biological characteristics of agricultural animals always evolve under the synergistic effect of natural selection and artificial selection, and the two breeds of cattle, Yunling and Leiqiong, are no exception. Yunling cattle were bred by cross breeding, and their growth and development were greatly improved, which was mainly due to the favorable accumulation of gene frequency by artificial selection (15 generations). The discussion part of the manuscript focuses on the analysis of functional genes. If the association of functional genes with selection and genetic background (parents of cross breeding) is properly analyzed in the discussion, it will be more beneficial for readers' interest and understanding of the results.
The following suggestion may be helpful to improve the quality of this manuscript.
Abstract:"SLC38A1, SLC38A3, DGKH, GNB4, GNAQ, ESR1, NPY, and GAL are new candidates in the pituitary gland for regulating ......", which is not accurate. None of these genes are new candidate genes for regulating the growth of cattle by regulating the secretion of growth-related hormones in the pituitary gland.
Author Response
Please see the attached PDF file.

Round 2
Reviewer 2 Report
>P5L190: R2 generally means the coefficient of determination. Did the author want to show the value of R2? The authors must check this point.
Response: Thanks for your advice. In fact, we have carefully discussed on this point these days and we think that the higher R2 value mean a stronger linear relationship between the genes’ expression levels detected by FPKM and RT-qPCR. In this case, if the expression levels of genes detected by RT-qPCR were low, then the FPKM value would be low, and if the expression levels of genes detected by RT-qPCR were high, then the FPKM value would be high too. So, the FPKM value can roughly represents the true expression of genes. Otherwise, if the R2 is low, which reflects the FPKM value could not represent the true expression of genes well, so we leave R2 in this manuscript to illustrate the accuracy of the sequencing data, thanks again.
I simply wanted to ask the authors to check whether R2 was used as correlation coefficient or coefficient of determination in this manuscript.
If the authors used R2 as correlation coefficient, it seems to be better to use ρ.
This manuscript is a resubmission of an earlier submission. The following is a list of the peer review reports and author responses from that submission.
Round 1
Reviewer 1 Report
The important biological characteristics of agricultural animals always evolve under the synergistic effect of natural selection and artificial selection, and the two breeds of cattle, Yunling and Leiqiong, are no exception. One of the purposes of this manuscript may be to show that the growth differences between Yunling and Leiqiong are related to the expression abundance of related hormones. Although there have been many studies on the association of the growth with hormones, readers still hope for some new findings in this manuscript.
The transcriptome analysis can reveal the difference of gene expression level, but it is easy to be affected by the background gene, which makes the result false positive. It is also very important for this manuscript how to eliminate the false and retain the true, including expanding the sample size.
The following suggestions may be helpful to improve the quality of this manuscript.
Introduction: The origin, ecological environment, growth and development characteristics of Yunling cattle and Leiqiong cattle should be described in detail (, which is the basis of the theoretical hypothesis of this manuscript).
Materials and Methods
Add data analysis methods, especially statistical processing methods of measured data.
Add the analysis on the 6 hormones from the results in RNA-seq
Line 85: The nutritional level, management, and growth (daily gain, for six months) of 6 test cattle samples should be described in detail.
Line 92: Simmental cattle should be not Leiqiong cattle. Need author's explanation. What's the meaning of the same word on line 343 and Line 383.
Line 100: " at 3000 rpm for 30 minutes ", should been shown the"...×g".
Line 230-231: The expression level of the gene was different from that of FRMP, fragments per kilobase per million. The sentence needs to be improved.
Results
Line 187: " in Figure 8a ", should show the Figure 8a in the manuscript.
Line 211: "the co-expressed and uniquely expressed genes", in which the " co-expressed " is difficult to understand.
Discussion
Add the discussion on association of the different expression genes with the growth of two cattle breeds, and give some suggestions.
Conclusion
The conclusion section needs to be revised to make it short and clear.
Reviewer 2 Report
In the manuscript entitled “Comparative Transcriptomic Analysis of Pituitary gland between Cattle Breeds Differing in Growth: Yunling Cattle and Leiqiong Cattle” (animals-831970), the authors revealed a possible regulation mechanism of pituitary on beef cattle growth via intervening the secretion of growth-related hormones in the blood and 9 candidate genes were detected to participate in this process. The authors reported new and attractive data on the transcriptome of pituitary gland and the secretion levels of growth-related hormones in blood of cattle at different growth levels. The study is of interest and convincing. This manuscript is suitable to be considered for publication but some details still need to be improved. The following are some suggestions for modification:
Title:
Line 3: Capitalize the first letter of these words, ‘gland’ and ‘between’;
Abstract:
Line 33: Please pay attention to tense issues;
Line 39: All the genes should be italicized at scientific manuscript.
Introduction:
Line 67-70: Please give a reference on the basic situation of Leiqiong cattle;
Materials and Methods:
Line 92-95: The statement of the sampling process is not coherent enough, please improve the English editing.
Line 105-106: Please give more details on the “Bovine specific ELISA kits”, such as the cat number of them.
Line 168: Please unify the writing of p value in this text, ‘P’ needs to be italic and lower case. (The same problem in the note of figures and tables)
Results:
Line 252: Why one of the identification criteria for DEGs in this article, log2|FoldChange| need to more than 3? As we know that this standard is generally greater than 2 which is enough for DEGs detecting and please give an explanation on it.
Discussion:
Line 345-350: Please unify the tense of these sentences.
Line 491-420: Although the reference 61 can explain the issues discussed well, but this document published in 1985 and a little old to be the reference. Please replace a newer document to illustrate the relationship between NPY and growth hormone.
Conclusion:
This section is not very close to the topic of this article, it is recommended to modify it.
Line 431-432: Maybe the conclusion “a comprehensive exploration of the genes expressed in the pituitary gland of cattle” cannot cover the topic of this article.
Line 432-433: There is no result or discussion on the five gene “POMC, GH, CGA, NNAT, and LHB”, how get this conclusion?
Line 435-437: In the elsewhere in this article, there are 9 genes were identified could affect the growth, not just 8. And the gene names are different, please check it.
Reviewer 3 Report
This article might contain interesting insight about the biological mechanism of the growth and development in cattle. However, I have found many points to be fixed, which made me lose my motivation. Therefore, I recommend that the manuscript is once rejected, and this might be resubmitted after careful checking by the authors.
Some of the points I found are described below.
1) P1L27: “Hypothalamic-Pituitary-Thyroid Axis (HPT) axis” → “Hypothalamic-Pituitary-Thyroid (HPT) axis”.
2) P1L39: The gene “CAL” should be in italic letter.
3) P2L45: impact, adjust, improve → impacts, adjusts, improves.
4) P2L86: Why the authors used only bulls? How about the influence of sex (male, female, or castrated) in this study?
5) P3L92: Did the authors use Simmental individuals in this study?
6) P3L108: What did this sentence mean?
7) P9L110: “The coefficient of variance” → “The coefficient of variation”.
8) P4L142: Why the authors used FPKM, but not TPM?
9) P4L143: “DEGs” → “differentially expressed genes (DEGs)”.
10) P5L187: Figure 8 did not exist in the manuscript.
There could be more points to be fixed than I have listed above. But I did not have time to find them all, and it is authors’ responsibility.
Reviewer 4 Report
General comments
- The English needs to be checked for this manuscript as there are grammatical and language errors throughout.
- Need more information in relation to how the growth rates between the two breeds differ, is one early maturing and the other late maturing or what is the difference?
- This paper needs more details in relation to the animals used including: 1) the dietary intake and growth rate of the cattle prior to starting the trial; 2) the age of all animals used; 3) the composition of the diet that the cattle received prior to the trial; 4) the growth rates from all animals during the trial; 5) the final live-weights for all animals at slaughter. It is not feasible to draw conclusions on the results presented without fully knowing the experimental animal model details and as that is involved in growth, details of the actual growth of the animals is needed.
- At slaughter, were any other body/organ measurements made? For example the thyoid gland or maybe the liver? These could be assessed relative to live-weight to determine a potential relationship with the hormones targeted. In particular the thyroid hormones which are centrally involved in metabolic rate.
- More specific information as to how the results from this study will actually be used should also be included.
- Methods section needs to clearly specify what was done, based on the results section there are some details missing from the methods.
- Gene ontology results, e.g. pathways and the involved genes should be included in supplementary results
Specific comments
Line 17: in what way is the growth between the two breeds extreme?
Line 38/39: are these really new candidates for growth, as some of these genes are already known to be involved in growth. The diet of the cattle/stage of development could also be impacting on these genes as well as the hormones assayed
Line 66-67: the difference in growth between these two breeds should be included here
Line 86: n=3 is very low for this type of experiment, how can the authors be sure that the results presented are reflective of the breed type and not just of the small number of individual animals used for the study?
Line 86: ‘were randomly selected and raised’ how were they randomly selected? At what age were they randomly selected?
Line 88: there is no information in relation to the diets offered, these details need to be included. Also the prior growth and dietary intake of the animals needs to be outlined as early life growth and intake can impact on growth potential at subsequent developmental stages. Also is the difference in growth between these two breeds taken into account in relation to the diets they were offered prior to the trial at 18 months?
Line 89: how were they slaughtered? What diets were they given up to slaughter? How long after they had last been fed were they slaughtered?
Why are Leiqiong used on line 86 but then it is simmental for other parts of the manuscript?
Line 92: given the large difference in body weight at slaughter, details on prior growth and dietary intake before the trial are required
Line 98: did the cattle have access to feed throughout the day, or had feed been removed from them until a blood sample was taken? All details in relation to feeding and recording growth of the animals used needs to be included within the manuscript
Line 114: was the entire pituitary used or just the anterior section?
Line 118: how did you determine that samples were of good enough quality, usually the RIN value derived from Bioanalyser or TapeStation measurement is used to determine RNA quality.
Line 140: how were reads determined to be high-quality or not? Were sequencing indexing adapters removed from sequencing reads?
Line 146: the fold change cut-off seems very stringent, more typically RNAseq studies report Log2-fold change>0.58, which equates to an actual fold change of 1.5.
Line 148: It is not clear that qPCR was run on the same biological samples as RNAseq?
Line 161: was only one reference gene tested, the MIQE guidelines for qPCR state to test a number of reference genes and only use the most stably expressed ones.
Lines 173-178: is this the statistical analysis between the hormones measured? It is not clear what this section is in relation to?
Lines 189-190: based on the hormone results and the weight difference at the time of slaughter, do the results not suggest that the Yunling cattle were still in a growing phase whereas the Leiqiong cattle were at a later stage of growth and not growing as much. The higher insulin in the latter may also be suggestive of this.
Line 210: did you evaluate what the breed specific genes were for each breed type and how they differ between the breeds?
Lines 223-225: does this imply that 6 reference genes were tested and not just GAPDH as described in the methods?
Line 243: I may have missed it, but where are the statistical methods outlined for the correlation?
Line 246: is there any reasoning as to why the Yunling group aren’t as close on the PCA plot, details on final live-weight, dietary intake, and average daily gain for all animals used may rationalise this image. Notwithstanding that, those types of details should be included within the manuscript.
Line 332: were bulls not used for this study?
Line 334-335: as bodyweight and age increase, the body becomes more insulin resistant and this may be why insulin is higher in the Leiqiong cattle.
Line 343: is the simmental work a different study and incorrectly used in the methods section?
Line 403: what are the 6 pathways?
